# Detection of the Core Bacteria in Colostrum and Their Association with the Rectal Microbiota and with Milk Composition in Two Dairy Cow Farms

**DOI:** 10.3390/ani11123363

**Published:** 2021-11-24

**Authors:** Bin Chen, Guangfu Tang, Weiqing Guo, Jie Lei, Junhu Yao, Xiurong Xu

**Affiliations:** College of Animal Science and Technology, Northwest A & F University, Xianyang 712100, China; chenbinnwafu@163.com (B.C.); tangguangfu@nwafu.edu.cn (G.T.); guoweiqing@nwafu.edu.cn (W.G.); leijie@nwafu.edu.cn (J.L.); yaojunhu2008@nwsuaf.edu.cn (J.Y.)

**Keywords:** Holstein cows, colostrum, bacterial composition, pathogenic bacterium, somatic cell count

## Abstract

**Simple Summary:**

In order to provide information on developing probiotics for newborn calves, this research detected the bacterial composition in colostrum and rectal feces of healthy Holstein cows from two dairy farms. Our results found several core bacterial species and some core genus and families in colostrum. About half of the OTUs detected in colostrum were found in the rectal content including some strictly anaerobic bacteria. In addition, some well-known intestinal beneficial bacteria including *Lactobacillus plantarum* and *Bacillus subtilis* were present in cow colostrum. Our results confirm that colostrum provides intestinal probiotics for calves. Furthermore, we might be able to develop new probiotics for calves according to the core symbiotic genus or families in colostrum.

**Abstract:**

As one of the pioneer bacterial sources of intestinal microbiota, the information of bacterial composition in colostrum might provide a reference for developing specific probiotics for newborn calves, especially calves fed with pasteurized milk. The present study aimed to detect the core bacteria at different taxonomic levels and the common beneficial ones in colostrum by analyzing the bacterial composition in 34 colostrum samples of healthy cows selected from two dairy farms. The results of the further analysis showed that the bacterial composition in the colostrum of the two dairy farms was different, but their four most dominant phyla were the same including *Firmicutes*, *Bacteroidetes*, *Proteobacteria*, and *Actinobacteria.* The microbiome of all colostrum samples shared ten core operational taxonomic units (OTUs), 21 core genera, and 34 core families, and most of them had no difference in relative abundance between the two farms. The ten core OTUs did not belong to the identified commensal bacteria and have not been detected by previous study. However, several core genera found in our study were also identified as core genus in a previous study. Some well-known beneficial and pathogenic bacteria including *Lactobacillus plantarum*, *Bacillus subtilis*, *Acinetobacter lwoffii*, and *Streptococcus pneumoniae* were present in the colostrum of healthy cows. However, none had a correlation with the number of somatic cell count (SCC), but the core genera *Nubella* and *Brevundinimas* and the core families *Methylobacteriaceae* and *Caulobacteraceae* positively correlated with the number of SCC. The genus *Staphylococcus*, *Pseudomonas*, and *Chryseobacterium* in colostrum had a positive correlation with each other, while the probiotics *unidentified*-*Bacteroidales*-*S24*-*7*-*group* had a negative correlation with *Pseudomonas* and *Chryseobacterium*. In addition, more than 50% bacterial OTUs in colostrum were detected in the rectal content including some strictly anaerobic bacteria that are generally present in the intestine and rumen. However, of the top 30 commonly shared bacterial genera in the colostrum and rectal feces, no genus in colostrum was positively correlated with that same genus in rectal feces. In conclusion, the bacterial composition of colostrum microbiota is greatly influenced by external factors and individuals. There were several core OTUs, and some core genus and families in the colostrum samples. Colostrum from healthy cows contained both beneficial and pathogenic bacteria and shared many common bacteria with rectal content including some gastrointestinal anaerobes.

## 1. Introduction

Colostrum is the milk secreted in the first few days after birth and characterized by high protein and antibody content. Colostrum contains diverse nutritional components including casein, lactose, fat, vitamins and minerals, and different bioactive components such as functional proteins, miRNAs, other immunomodulatory factors, and immune cells [1,2]. These bioactive components protect the neonate by inhibiting pathogens and other postpartum environmental challenges [3], or by regulating the development of the intestinal microbiota and immune system [3,4].

Except for functional components above-mentioned, more and more studies have proven that there are abundant bacteria in both the colostrum and regular milk of healthy mothers [5]. As the pioneer microbial source of newborns, the bacteria in colostrum may affect the development of the intestinal microbiota and the bacteria-induced development of the intestinal immune system [6]. However, both colostrum and regular milk are pasteurized before being fed to calves in order to protect them from pathogenic infection in many dairy farms in China. The pasteurization process inactivates some of the bioactive components and kills most of the bacteria including the beneficial ones, which might consequently inhibit the normal development of intestinal microbiota and beneficial bacteria-induced development of the intestinal immune system. The possible negative effects caused by pasteurization might be minimized if appropriate beneficial bacteria are added to the pasteurized colostrum and regular milk.

The application of probiotics in neonatal calves has been investigated [7,8]. The typical bacterial composition in the colostrum or regular milk is determined primarily by the evolution of the host and may exhibit host specificity, and therefore provide information on how to select an appropriate bacterium or bacterial combination as the probiotics for newborn calves. A previous study detected bacterial composition in the colostrum of cows with or without mastitis or antibiotics [9,10]. However, the bacterial community in colostrum or regular milk is complicated and might be affected by some factors including the geographical location [11] and diet composition. At the same time, there might be core bacteria in healthy colostrum or regular milk that are not affected by external factors and play important roles in the development of intestinal microbiota. The present study aims to detect these core bacteria at different taxonomic levels in colostrum by analyzing the bacterial community structure of colostrum from two dairy farms based on high throughput sequencing of 16S rDNA and comparing the core bacteria with those detected by Lima et al. [9]. More and more research suggest that the intestinal bacteria can transfer to the breast and contribute to the origination of bacteria in colostrum [12,13,14]. Species in the rectal feces are partly represented in the intestine. Therefore, the rectal feces samples of the same cows were also collected after calving to investigate the intestinal microbiota and analyze its relationship with the bacteria in colostrum. The correlation of the detected bacteria and the SCC number in colostrum was also investigated.

## 2. Material and Methods

### 2.1. Study Design and Study Population

We used an observational study design in which colostrum and rectal feces samples were aseptically collected from healthy cows in two commercial dairy farms: the X dairy farm and H dairy farm in Shaanxi Province. The two farms are located in different regions of Shaanxi Province. A total of 39 healthy pregnant multiparous Holstein cows in these two farms (17 from X Dairy Farm was selected in September to October, 22 from H Dairy Farm, and colostrum was selected from September to October) were identified during their last two weeks of pregnancy and deemed eligible for gradual enrollment in the sampling procedure based on the following criteria [15]: the healthy appearance of the four quarters (no visible sign of clinical mastitis such as swelling or redness, and devoid of anatomically damaged teat ends) and no incidence of clinical mastitis during the last 60 days of the previous lactation. At the end of the previous lactation cycle, all cows were subjected to blanket dry cow therapy using internal teat sealant. During the late-pregnancy period until three days post-calving, cows were housed in designated transition pens bedded with fresh and dry straw. Afterward, all cows were transferred to free-stall pens bedded every other day with treated recycled bedding material. Following parturition, cows were milked three times a day. No probiotic, prebiotic, or antibiotic was supplemented to the selected cows. The cows in the same farms were fed with the same diet, and the diet formula (not allowed to be published) at the two farms was different.

### 2.2. Collection of Colostrum and Feces Samples

Fresh colostrum samples from the healthy selected cows (*n* = 35, 16 from X dairy farm, 19 from H dairy farm) were collected within 6 h after calving. On the day of sampling, the four quarters of each cow were rechecked to make sure the cow had no clinical signs of mastitis (swelling or redness). Prior to sampling, the teats were washed twice using sterilized ultrapure water, and then disinfected using 0.5% iodine pre-dip solution and 75% alcohol before being scrubbed with sterilized cotton pads. The last scrubbing cotton pads were collected to investigate the existence of bacteria on the surface of the treated teats. To minimize the chances of sample contamination from bacteria colonizing the teat canal and to check for the abnormal appearance of colostrum (i.e., watery secretions, presence of blood, flake, or abnormal color), the first four streams from all quarters of each selected cow were discarded. Cows diagnosed with clinical mastitis at any sampling time were excluded from the study, and one cow from the X dairy farm and three cows from the H dairy farm were excluded after the first selection due to clinical signs of mastitis. The colostrum samples were then collected, and about 50 mL of colostrum from each cow was quickly put into a 50 mL sterile centrifuge tube and stored in liquid nitrogen, then transferred to a −80 °C refrigerator in the laboratory for further processing. Another 10 mL of colostrum from each cow was put into a tube containing the preservative and stored in an ice box to determine the somatic cell counts of fat, protein, and lactose contents in the colostrum sample (MilkoScan Type FT120, Foss Electric, Hillerød, Denmark). Calves were separated from cows soon after birth to prevent suckling. The rectal feces samples of the same cows were collected from their rectum within 6 h after calving and temporarily stored at liquid nitrogen and transferred to the −80 °C refrigerator in the laboratory until analysis.

### 2.3. Extraction of Genomic DNA in the Collected Colostrum and Fecal Samples

The bacterial genomic DNA of the colostrum and rectal feces samples, and of the collected last scrubbing cotton pads was extracted using the CTAB (cetyltrimethylammonium bromide) method [16]. The extracted DNA was dissolved in a sterile TE buffer, and its concentration was detected by a micro nucleic acid analyzer (Nanodrop 2000, Thermo Fisher Scientific, Wilmington, DE, USA) and agarose gel electrophoresis. DNA samples that met the quality requirements were sent to Beijing Allwegene Gene Technology Co. Ltd. for 16S rDNA high-throughput sequencing analysis.

### 2.4. The 16S rDNA High-Throughput Sequencing of Microflora in Colostrum and Feces

The V3–V4 region of the bacterial 16S rRNA gene was amplified from the metagenomic DNA of all samples and sequenced by using an Illumina HiSeq2500 high-throughput sequencing system. The oligonucleotide primers for high-throughput sequencing were 338F and 806R [17]. Quality filtering on raw tags was performed using specific filtering conditions to obtain high-quality clean tags with QIIME software (V1.7.0) [18]. All sample libraries were rarefied to an equal depth of 24,780 sequences using QIIME. The operational taxonomic units (OTUs) were based on ≥97% sequence similarity [19]. Alpha and beta diversity were calculated on the basis of the de novo taxonomic tree constructed by the representative chimera-checked OTUs set using FastTree. The Shannon–Wiener, Chao1, and rarefaction estimators were performed to evaluate the sequencing depth and biodiversity richness. To assess the microbiota structure of different samples, the principal coordinate analysis (PCA) was applied using the weighted and unweighted UniFrac distances derived from the phylogenetic tree.

### 2.5. Statistical Analyses

All experimental data were analyzed with the R software (v3.1.3). Statistical significant differences were tested based on a Mann–Whitney test in a pairwise manner. *p*-values below 0.05 were considered statistically significant. To adjust for falsely rejected null hypotheses, the false discovery rate (FDR) was calculated by comparing colostrum and fecal bacteria proportions at each phylogenetic level separately. The correlation between fecal bacteria and milk components, SCC, and other measured parameters, or between the bacterial genus in colostrum and that same genus in rectal feces were represented by the Spearman rank correlation coefficient and visualized by heatmap in R using the “heatmap” package. The graphic presentations were generated by Graph Pad Prism v.8.

## 3. Results

### 3.1. General DNA Sequencing Observations

The bacterial community structure in the colostrum of cow was investigated by sampling colostrum from two different farms. In the current study, a total of 1,709,820 high-quality sequences were used for analyzing the 34 colostrum samples (the genomic DNA of one colostrum sample from H dairy farm failed to amplify the PCR product). For this reason, in the subsequent analysis, 16 samples from dairy farm H and 18 samples from dairy farm X were used for statistical analysis concerned with colostrum. The total number of unique and classifiable representative OTU sequences for bacteria was 7665. The rarefaction curve of all samples was performed at the OTU level (Appendix A), and both of them indicated that the sampling effort had sufficient sequences to detect the majority of bacterial diversity. The DNA concentration extracted from the collected cotton pads was too low to sequence.

### 3.2. Alpha and Beta Diversities of Bacterial Composition in Cow Colostrum of the Two Dairy Farms

The estimators of community diversity (Shannon) and richness (Chao1) are shown in Figure 1. The richness between colostrum samples from the two farms was similar (*p* > 0.05), but the Shannon index of colostrum in dairy farm H was significantly higher than that in dairy farm X (*p* < 0.05).

The community similarity at the phylum level was evaluated by constructing the non-metric multidimensional scaling (NMDS) ordination plot of Bray–Curtis community dissimilarities based on OTUs from the 16S rRNA gene sequences (Figure 2). The bacterial community profiles of most samples from dairy farm X clustered to the left of the NMDS plot, while that from dairy farm H clustered to the right. Furthermore, the bacterial community profile of the colostrum samples from the same farm was relatively divergent, especially in dairy farm H.

Analysis of similarities (Anosim) provides a way to test statistically whether there is a significant difference between two or more groups of sampling units. As shown in Figure 3, the result of the Anosim analysis suggested that the colostrum bacterial composition difference between the two farms was more prominent than that among cows of the same farm (R = 0.318, *p* = 0.001).

### 3.3. Taxonomic Analysis of Bacterial Community Structure in Colostrum of the Two Farms

The bacterial community structure of colostrum was analyzed at the phylum to genus level. The four most common bacterial phyla in colostrum collected from both farms were *Proteobacteria, Firmicutes*, *Bacteroidetes,* and *Actinobacteria* (Figure 4A). There were significant differences in the microbiota structure of colostrum at the genus level between the two farms (Figure 4B). The first abundant genus in most colostrum samples of dairy farm X was *Psychrobacter*, which was not abundant or absent in most of the samples from dairy farm H. While the most abundant genus in samples from dairy farm H was an unidentified genus (belonged to *Rhodospirillaceae*), which was also present in high abundance in all samples of dairy farm X. The top 20 genera in the colostrum collected from the two farms included genera *Staphylococcus*, *Pseudomonas*, *Escherichia,* and *Chryseobacterium*.

### 3.4. Analysis of the Core Bacteria in the Collected Colostrum Samples

The core microbiome was defined as the set of microbial organisms persistently present in all samples evaluated, regardless of the farm, parity, and sampling time in the present study. There were only 10 core OTUs (Table 1), 21 core genera (Table 2), and 34 core families (Appendix A) in 100% of the collected colostrum samples. The relative abundance of these ten core OTUs was higher than 0.1%, and none of them showed an obvious difference in relative abundance between the two farms (*p* > 0.05). The average relative abundance of OTU60, which belonged to genus *Anoxybacillus*, was the dominant OTU, but the individual abundance varied greatly from sample to sample (from 0.01% to 39.87%). Of the 21 core genera, only three had relative abundance difference between the two farms including genus *Paracoccus* (*p* = 0.031), *Pseudomonas* (*p* = 0.032), and *Christensenellaceae_R-7_group* (*p* = 0.010).

### 3.5. Analysis of Commensal Probiotics and Conditional Pathogen in the Colostrum Samples

There were some commensal beneficial bacteria in the collected samples (Appendix A) including *Bacillus circulans* (present in 75% of the collected samples), *Lactobacillus plantarum* (75%), and *Bacillus subtilis* (55%), but all had pretty low relative abundance (data not shown). Meanwhile, pathogens *Acinetobacter lwoffii* and *Streptococcus pneumoniae* were present in 100% and 94% of the collected samples. Some other well-known conditional pathogens including *Delftia tsuruhatensis* (91%), *Escherichia coli* (68%)*,* and *Pseudomonas aeruginosa* (50%) were also detected in the collected colostrum samples.

### 3.6. Correlations among the Different Genera in Colostrum

As shown in Figure 5, some bacteria in colostrum interacted with each other, either positively or negatively. The genus *Staphylococcus* had a strong positive correlation with *Pseudomonas* and *Chryseobacterium*, and the latter two were also positively correlated with each other. The *unidentified*-*Bacteroidales*-*S24*-*7*-*group* had a negative correlation with *Pseudomonas* and *Chryseobacterium*, and a positive correlation with several genera including *Christensenellaceae*-*R*-*7*-*group*, *unidentified*-*Clostridiales*-*VadinBB60*-*group*, *unidentified-Bacteroidales-RF16-group,* and *Treponema-2*.

### 3.7. Association of the Bacterial Composition in Colostrum and Rectal Feces

The Venn figure of the colostrum and the feces samples of the same dairy farm are presented in Figure 6). There were 3549 consensus OTUs between the colostrum and feces samples of NH dairy farm and 3335 consensus OTUs between the colostrum and feces samples of NX dairy farm. We also assessed the correlations between the relative abundances of the top 30 commonly shared bacterial genera in the colostrum and rectal feces. However, none of these genera in colostrum was significantly positively correlated with that same genus in rectal feces (*p* > 0.05) (Figure 7). In contrast, negative colostrum–gut correlation was strong for *Romboutsia* (R = −0.378, *p* < 0.001). In addition, positive or negative correlation were detected between some bacterial genera in colostrum and other bacterial genera in rectal feces. For example, *Staphylococcus* in colostrum had a markedly positive correlation with *Fastidiosipila* in rectal feces. Two core OTUs were found in all of the collected colostrum and rectal feces samples, and both belonged to *unidentified_Ruminococcaceae_UCG-005* and *Romboutsia*, respectively.

The NMDS plot showed distinct clusters for colostrum and feces (Appendix A), the bacterial community profile of colostrum grouped to the left of the NMDS plot and that of feces clustered to the right. The bacterial community profile of the colostrum samples was relatively divergent, while all of the rectal feces samples closely gathered and overlapped in the right. However, the rectal feces samples were clustered according to farm when the NMDS plot was constructed by evaluating the bacterial community similarity of feces samples only.

### 3.8. The Correlation between Bacteria and SCC or the Main Composition in Colostrum

The correlation between colostrum microbial structure and somatic cells and milk composition were analyzed. None of the top 20 genera had an obvious positive or negative correlation with the number of SCC (somatic cell count) in colostrum (Appendix A), and the relative abundance of well-known pathogenic or beneficial bacteria had no obvious correlation with the number of SCC in colostrum (*p* > 0.05) (Appendix A). The core genera *Nubella* (R = 0.419, *p* = 0.0136) and *Brevundinimas* (R = 0.4105, *p* = 0.0159), and core families *Methylobacteriaceae* (R = 0.388, *p* = 0.023) and *Caulobacteraceae* (R = 0.388, *p* = 0.023) had a positive correlation with the number of SCC in the collected colostrum samples (Appendix A). The family of *Staphylococcaceae* or the genus *Staphylococcus* or the detected species belonging to *Staphylococcus* all had no obvious relation with SCC. Other components in colostrum were correlated with specific bacteria in colostrum, among which the lactose content was positively correlated with *Psychrobacter* and *Aequorivita* but negatively correlated with the *Christensenellaceae R-7 group*; the fat content positively correlated with *Bacteroides*; and urea nitrogen content negatively correlated with *Staphylococcus.*

## 4. Discussion

### 4.1. The Bacterial Compositions in the Colostrum of the Two Dairy Farms Were Different but Relatively Stable at Higher Taxonomic Levels

The Chao1 and Shannon index indicated that the bacterial composition in the colostrum of cow was rich and diverse. As one of the first sources of intestinal bacteria provided for newborn calves, the bacteria in colostrum may play an important role in the early establishment of the intestinal microbiota and the development of a bacteria-induced intestinal immune system. However, many dairy farms in China feed calves with pasteurized milk in order to prevent intestinal pathogen infection. Although the pasteurization process reduces the rate of intestinal infection, it may also interfere with the establishment of the intestinal microbiota of calves. This possible adverse effect brought by pasteurization might be alleviated through supplementation with appropriate probiotics in pasteurized milk. Therefore, understanding the bacterial composition of colostrum could help us develop and select the appropriate probiotics for calves.

The bacterial composition of colostrum should be different at different dairy farms due to different environment, management, and diet. As expected the differences in the Shannon index and the dendrogram of bacterial composition, the bacterial community profiles between the two farms confirmed our conjecture. Although we compared the colostrum microbiota of only two dairy farms, the obvious differences between them were sufficient to indicate that colostrum microbiota was influenced by farm. The NMDS plot of bacterial community suggested that there was great individual difference in the bacterial structure of colostrum. Further study should be conducted to investigate the main factor contributing to the difference in the bacterial composition of colostrum between the farms and individuals.

There were only ten core OTUs in all of our collected colostrum samples. Although there were significant differences in the structure of colostrum microbiota between the two farms, there was no obvious difference in the relative abundance of the ten core OTUs between the two farms, suggesting that they might be conserved OTUs in colostrum. However, these ten OTUs did not match the core OTUs detected by Lima et al. [9]. However, several of the 21 core genera detected in the present study were also core genera in cow colostrum in two other studies [9,20] including *Bacillus, Bacteroides, Staphylococcus*, *Acinetobacter*, and *Pseudomonas*. Furthermore, the four most dominant phyla in colostrum from the two farms were the same and consistent with the published study of Lima et al. [9].

An important objective of this study was to provide a reference for developing and selecting probiotics for newborn calves based on the core OTUs investigated in colostrum samples collected from different farms. According to the present and previous results [9], it can be inferred that the structure of the core bacteria at the species level in colostrum is unstable but relatively stable at the genus level, especially at the phylum level. Therefore, it remains to be studied whether the farm factor should be considered in the probiotic selection for newborn calves, and whether genera composition instead of species composition should be considered first when supplementing with probiotics in pasteurized milk.

### 4.2. Particular Bacteria Existed in the Collected Colostrum and Their Possible Function

None of the ten detected core bacteria belonged to the commercially applied probiotics. However, as Lyons et al. [21] pointed out, research looking at unconventional bacterial species may shed new light on the development of probiotics. In fact, it is speculated from some studies that certain bacteria in milk may play important roles in the gut health of neonates. For example, studies have shown that some species of *Flavobacterium* and *Pedobacter* in milk could use lactose to produce epilactose [22], a potential prebiotic of *Bifidobacteria* and *Lactobacilli* [23]. In addition, bacteria belonging to *Sphingobacteriaceae*, one dominant family in our collected colostrum, can produce sphingolipids, which have beneficial effects on gut health and immunity in infants [24]. It is worth investigating whether these bacteria use milk components in the mammary gland or intestinal tract to produce functional metabolites useful for calves.

The presence of many common intestinal symbiotic genera in all or part of the collected colostrum samples including *Bacteroides, Clostridium*, *Ruminococcus*, *Lactobacillus*, *Bacillus, Bifidobacterium,* and *Prevotella*, suggested that colostrum is one of the early sources of intestinal microbiota in newborn calves. Since some of the beneficial bacteria in colostrum are strictly anaerobic, they may be the main early source of such beneficial bacteria in the intestine of calves.

Pathogenic bacteria were detected in the colostrum of healthy cows. Of the ten core OTUs, two belonged to the genus *Acinetobacter* and one belonged to the genus *Chryseobacterium*. In addition, several pathogenic or opportunistic pathogen bacteria including *Streptococcus pneumoniae*, *Pseudomonas aeruginasa*, *Delftia tsuruhatensis* [25], *Stenotrophomonas maltophilia* [26], and *Escherichia coli* were present in most of the collected colostrum samples. Genus *Streptococcus, Acinetobacter*, *Escherichia,* and *Pseudomonas* were also detected in healthy colostrum samples in the study of Lima et al. [9]. Pathogenic bacteria in colostrum or regular milk are considered to be detrimental to newborns. However, recent studies have proven that animals exposed to a certain amount of pathogen have stronger resistance to subsequent infection [27]. In fact, maturation of the adaptive and innate immune systems results not only through vaccination, but also through other microbial exposures such as diet and the uterus [28,29]. Further study should be performed to detect whether early exposure of the newborn to certain pathogens through colostrum is a kind of early immune training.

*Staphylococcus, Pseudomonas,* and *Chryseobacterium*, three genera with pathogenic tendencies, had a strong positive correlation with each other in colostrum, but the beneficial bacterium *Bacteroidales*-*S24*-*7*-*group* [30] had a negative correlation with *Pseudomonas* and *Chryseobacterium*. The detected correlation among genus indicated a specific competitive relationship between beneficial bacteria and pathogenic ones in the colostrum of healthy cows.

### 4.3. Relation of Bacteria in Colostrum with Bacteria in Rectal Feces

More and more research suggest that the maternal intestine microbe is one of the origins of microorganisms in milk (Jost et al., 2014; Addis et al., 2016) [14]. Studies have suggested that some bacteria in the gut can be taken into the dendritic cells and macrophages, then transported to the mesenteric lymph nodes and further transfer to the distal organs such as the mammary gland [4,31]. More than 50% of the OTUs found in the colostrum were detected in rectal feces, and some strict anaerobes that commonly inhabit the gastrointestinal tract were present in the collected colostrum such as *Clostridium butyricum* and *Bacteroides fragilis*, the core OTUs belonged to anaerobic *Ruminococcaceae_UCG-005*, and several other genera of *Ruminococcaceae*. As strict anaerobes are present in colostrum, it is unlikely to come from the surface of nipples [9]. Therefore, the intestinal bacteria should be one origin of the strict anaerobes in the colostrum of cows. In turn, these strict anaerobes in colostrum should be an important early source of intestinal strict anaerobes in newborn calves. The fact that all of the top 30 commonly shared genera in colostrum had no significantly positive correlation with that same genus in rectal feces suggests that rectal microbiota is not the main sources of bacteria in colostrum. Therefore, correlation between microbiota in colostrum and that in other parts of the gastrointestinal tract should be further analyzed to investigate the origin of bacteria in the colostrum of cows.

### 4.4. Correlation between Bacteria and SCC or Other Compositions in Colostrum

The number of SCC is a primary indicator of mammary gland inflammation and mastitis caused by pathogenic infection. However, no significant correlation between the 20 top genera and the number of SCC in colostrum was investigated in the present study. Some other genera or families had a positive relation with the number of SCC in colostrum including genera *Nubella* and *Brevundinimas*, families *Methylobacteriaceae* and *Caulobacteraceae*, but none of them has been reported as an inducement of mastitis in cows. *Staphylococcus aureus* is one of the inducements of mastitis, but its relative abundance in the collected colostrum did not correlate with the number of SCC. Lima et al. [9] reported that mastitis had no significant impact on the microbial diversity and composition of multiparous cows.

It was reported that milk urea nitrogen is negatively correlated with the number of somatic cells [32] because some bacteria such as *Actinomycetes*, fungi, and species of *Staphylococcus* can produce urease to degrade urea in fresh milk [33,34]. This study supported the above conclusion by showing that the relative abundance of *Staphylococcus* was negatively correlated with the urea nitrogen content in colostrum.

## 5. Conclusions

In conclusion, the bacterial composition of colostrum from the two dairy farms differed, while shared some core bacteria. The ten core OTUs found in all of the collected colostrum samples did not belong to the well-known intestinal bacteria, and did not match with the core bacteria detected in a previous study [9]. However, some of the detected core genera and families belonged to the common intestinal symbiotic ones and were also identified as the core ones in other research. Colostrum from healthy cows contained both beneficial and pathogenic bacteria and shared many OTUs with rectal feces including some gastrointestinal anaerobic commensals.

## Figures and Tables

**Figure 1 animals-11-03363-f001:**
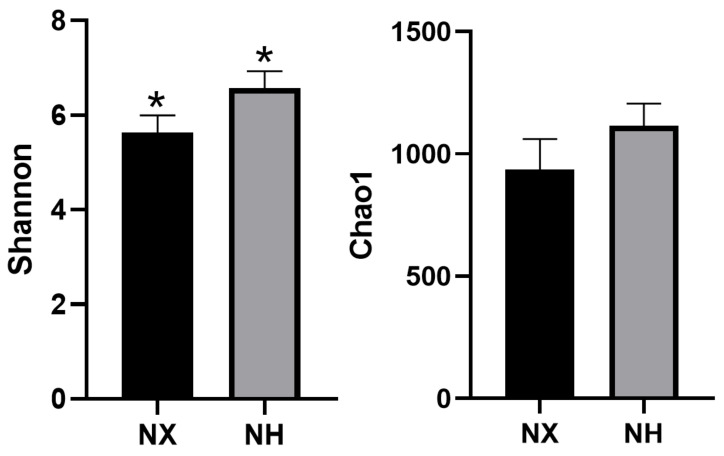
Comparison of colostrum microbial diversity and richness between dairy farm X and dairy farm H. Note: NX (*n* = 16), NH (*n* = 18) represent the colostrum samples from dairy farm X and dairy farm H, respectively; * the difference between the two farms was obviously different (*p* < 0.05).

**Figure 2 animals-11-03363-f002:**
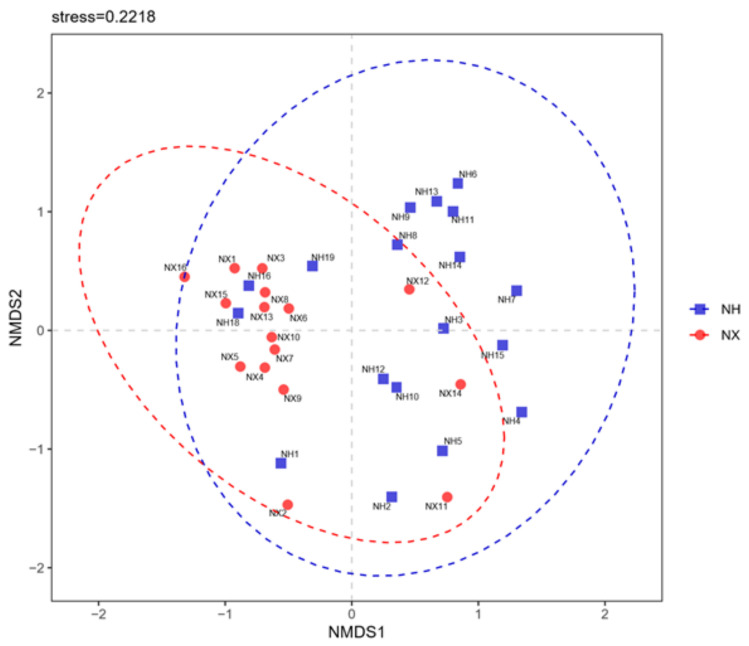
Non-metric multidimensional scaling of colostrum samples from the X and H dairy farms based on bacterial composition. Note: NX (*n* = 16) and NH (*n* = 18) represent the colostrum samples from dairy farm X and dairy farm H.

**Figure 3 animals-11-03363-f003:**
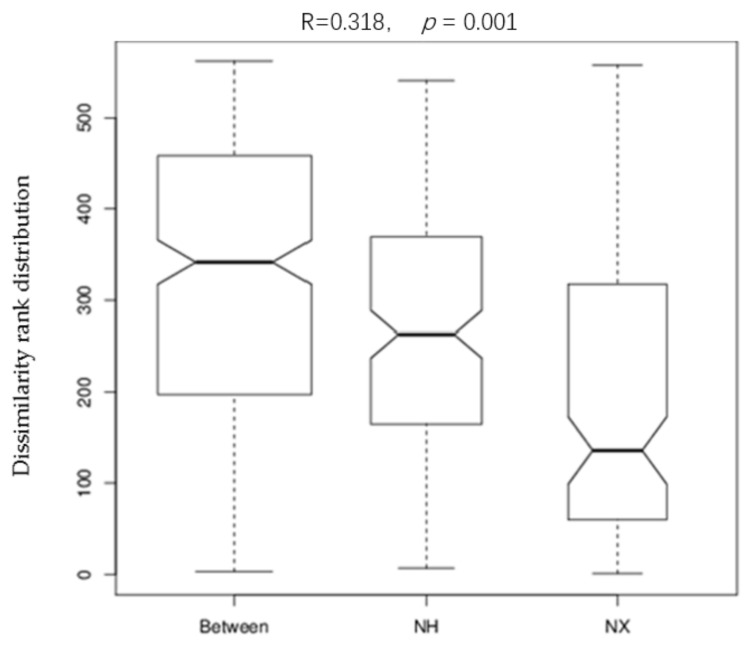
Anosim analysis of bacterial 16S rRNA gene sequence tags for colostrum samples from the X and H dairy farms. Note: NX (*n* = 16) and NH (*n* = 18) represented the colostrum samples from dairy farm X and dairy farm H.

**Figure 4 animals-11-03363-f004:**
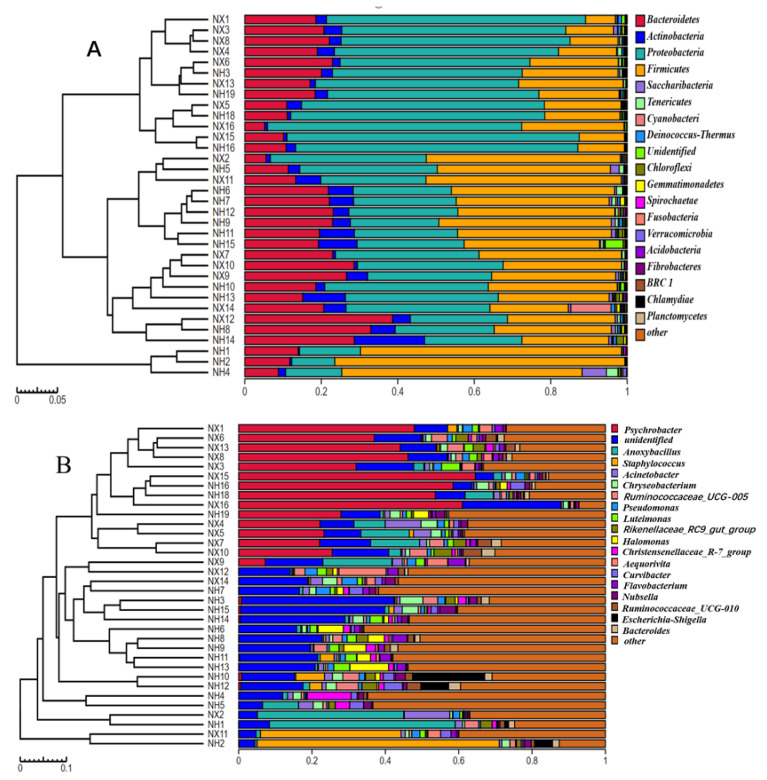
Dendrogram of bacterial composition analysis of colostrum from the X and H dairy farms. (**A**) Dendrogram of bacterial composition analysis of colostrum from the X and H dairy farms at the phyla level; (**B**) Dendrogram of bacterial composition analysis of colostrum from the X and H dairy farms at genus level. Note: NX (*n* = 16) and NH (*n* = 18) represent the colostrum samples from dairy farm X and dairy farm H.

**Figure 5 animals-11-03363-f005:**
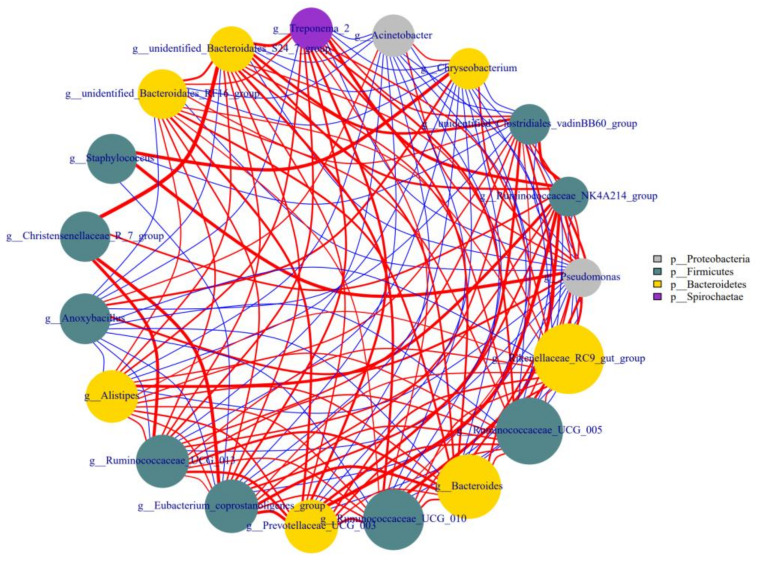
Network analysis of association of microbiota at the genus level in colostrum. Note: The number of colostrum samples for this analysis was 34.

**Figure 6 animals-11-03363-f006:**
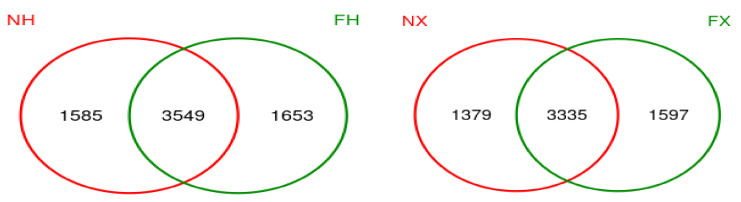
The Venn diagram of bacteria in the colostrum and feces samples of the same dairy farm. Note: NX (*n* = 16) and NH (*n* = 18) represent colostrum samples from dairy farm X and dairy farm H, FX (*n* = 16) and FH (*n* = 19) represent rectal feces samples from dairy farm X and dairy farm H.

**Figure 7 animals-11-03363-f007:**
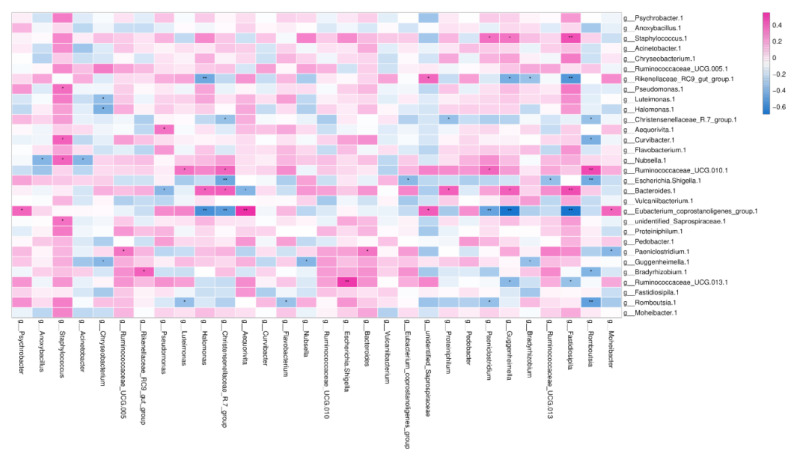
Spearman rank correlation between the relative abundances of the top 30 commonly shared bacterial genera in the colostrum and rectal feces. Note: The bacterial genera in milk and rectal feces were presented vertically and horizontally, respectively. g-genus; * *p* < 0.05; ** *p* < 0.001.

**Table 1 animals-11-03363-t001:** The core OTUs across 100% of the collected colostrum samples from the two farms and their relative abundance (%).

Number of the OTUs	Genus of the Core OTUs Belonged to	NX	NH	SEM	*p*-Value
36	*Acinetobacter*	0.32	0.28	0.14	0.787
40	*Acinetobacter*	1.32	0.62	0.70	0.323
58	*Curvibacter*	0.82	1.51	0.43	0.103
60	*Anoxybacillus*	6.54	4.42	3.88	0.587
1156	*Chryseobacterium*	1.46	1.76	0.38	0.441
1161	*Nubsella*	0.78	1.38	0.32	0.062
2182	*Unidentified genus*	0.18	0.50	0.17	0.065
2280	*Ruminococcaceae_UCG-005*	1.18	1.27	0.43	0.837
3169	*Sphingomonas*	0.37	0.58	0.18	0.256
3187	*Bradyrhizobium*	0.61	0.70	0.27	0.742

Note: NX (*n* = 16) and NH (*n* = 18) represent the colostrum samples from dairy farm X and dairy farm H, respectively.

**Table 2 animals-11-03363-t002:** The core genus across 100% of the collected colostrum samples from the two farms and their relative abundance (%).

Core Genus	NX	NH	SEM	*p*-Value
Unidentified genus	12.3	17.38	2.99	0.100
*Anoxybacillus*	6.55	4.42	3.88	0.587
*Staphylococcus*	2.78	4.71	0.45	0.669
*Acinetobacter*	2.65	1.41	0.86	0.192
*Pseudomonas*	1.84 ^a^	1.10 ^b^	0.33	0.032
*Ruminococcaceae_UCG-005*	1.69	1.88	0.60	0.762
*Luteimonas*	1.62	1.19	0.45	0.342
*Chryseobacterium*	1.59	2.13	0.48	0.269
*Flavobacterium*	1.32	1.04	0.39	0.462
*Rikenellaceae_RC9_gut_group*	1.30	1.41	0.47	0.829
*Pedobacter*	0.92	0.62	0.36	0.415
*Bacteroides*	0.89	1.02	0.35	0.701
*Curvibacter*	0.82	1.51	0.43	0.103
*Nubsella*	0.78	1.38	0.32	0.062
*Bacillus*	0.70	0.69	0.37	0.970
*Eubacterium_coprostanoligenes_group*	0.68	0.86	0.27	0.513
*Bradyrhizobium*	0.61	0.70	0.27	0.742
*Christensenellaceae_R-7_group*	0.50 ^b^	2.01 ^a^	0.48	0.031
*Sphingomonas*	0.40	0.66	0.20	0.208
*Brevundimonas*	0.36	0.42	0.17	0.727
*Paracoccus*	0.20 ^b^	0.48 ^a^	0.10	0.010

Note: ^ab^ Values in the same row with different superscripts differed (*p* < 0.05). NX (*n* = 16) and NH (*n* = 18) represent colostrum samples from dairy farm X and dairy farm H, respectively.

## Data Availability

None of the data were deposited in an official repository.

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
