# Peer review of "Detection of the Core Bacteria in Colostrum and Their Association with the Rectal Microbiota and with Milk Composition in Two Dairy Cow Farms"

_animals, 2021, doi:10.3390/ani11123363_

Round 1

Reviewer 1 Report

I read with interest the manuscript : Detection of the core bacreia in colostrum by analyzing colos-2 tral bacterial structure of two dairy farms and its association 3 with that in feces and milk composition from Chen et al. In this manuscript, the authors are presenting microbiota of the colostrum and feces in 2 dairy herds. Although I found merit to the manuscript I think there are many drawbacks and overinterpetation based on the findings. The authors are suggesting probiotic supplementation of colostrum after pasteurization in the abstract which is not supported by the current study findings. I think it is too prematurely in the current state of knowledge and the study has not this specific aim. I have also pointed specific points that need to be addressed concerning the specific study design and analysis. The authors should describe more thoroughly the 2 herds and management since the study results are strongly depending on these 2 herds and their management strategy (eg feeding, bedding …).

The english is poor and should be improved throughout the manuscript. I strongly suggest a reading by a native English speaker (I’ve corrected some errors but many are still there).

Specific comments

L31 : present vs previous?

L32 : consensus? Please modify

L43-44 : I do not agree with this statement. You did not prove that in this study.

L52-54: this sentence is difficult to read. The 2nd portion is not in adequation with the 1st part.

L59-66: pasteurization is a possible treatment of colostrum, I agree. However, pasteurization is not performed in the majority of dairy farms. Moreover, I would like to see reference concerning the eventual side-effects of pasteurization. Most articles seen in the short-mid term side effects are in favor of pasteurization (ie especially removing bacterial overload when colostrum is heavily contaminated) or with no difference between pasteurization and no thermic effects. The effect on microbiota (if existing) is not proven to my knowledge. There is no specific reference in this part (because evidence is lacking I think). It is well known that colostrum can be heavily contaminated (see seminal work from Fecteau et al., 2002). In this case, pasteurization would likely be more beneficial than nothing since the bacteria composition is not “normal” but heavily influenced by environmental (eg fecal) bacteria.

Fecteau, G., Baillargeon, P., Higgins, R., Paré, J., & Fortin, M. (2002). Bacterial contamination of colostrum fed to newborn calves in Québec dairy herds. The Canadian Veterinary Journal, 43(7), 523.

L72: bacteria

L72: mastitis vs mammitis

L85: Study design: due to the study goals and the fact that fecal microbiota is of interest, a minimal information should be given considering the herds characteristics including average milk production in the farm as well as feeding practices (depending on the feed program, this could impact rectal microbiota I think especially if some additive are used in the feed (eg rumensin, probiotic,…) and bedding characteristics (L100 you indicate this is recycled bedding, is this recycled after feces were heated?. Recycled bedding can be associated with high variability of processes and bacterial species. The recycling procedure should at least be developed minimally for the reader.

Figure 1: the figure resolution is poor and should be improved.

L182: I don’t agree that you’ve proved the cluster were distinct. The ellipses are not identical (I agree on the visual aspect). However, even if not represented we can easily guess that the 2 barycenters are included within the 95% CI ellipse. For this reason, I don’t think that there is a difference between herds (at least a relevant (stat sig) difference.

Figure 3: the axis legends should be changed rather than using the R default. + resolution problem

Figure 4: resolution problem here too. I’m not able to read the bacteria names. I is really an histogram or a dendrogram + stacked bar chart?

L246: define the % rather than saying “ a part “ which is not precise to be useful.

Figure 5: resolution is poor. In the legend we have no information on the meaning of red vs blue link as well as on the line thickness meaning. This should be indicated. Some long bacteria name are impossible to read especially when spread over the network. This should be changed by either reducing the length (abbreviation) or using a white box where to put the name.

L262: presented

L290-290: avoid including trends. Trends are misleading especially in a paper with many many many stat tests!!! (risk of false positive is there even if controlling (trying to) it)

Reviewer 2 Report

Brief summary:

This research article compares the bacterial composition of colostrum from healthy cows from two separate dairy farms. While some differences were identified, overall conserved bacterial profiles were detected between farms. Some genera of bacteria were additionally shared with paired fecal samples from cows. Collectively this information could be informative for commercial probiotic supplementation, and general understanding and characterization of microbiome impact and development.

Broad Comments:

English should be examined throughout: in the title I am not familiar with the term ‘bacreia’ and there is no milk composition work shown (only colostrum) thus I suggest rewording the title.  Also throughout the manuscript consider revising ‘bacterial structure’ to ‘bacterial composition’ or ‘bacterial profiles’ which I think more accurately describes the results presented.  

The study is well written, thoughtfully discussed, and data presented is well analyzed. However, the significance of the study is extremely limited. The study would be much more valuable and interesting if additional components were assessed such as more farms across a geologic region to develop bacterial composition consensus, between colostrum and milk samples, or between healthy versus mastic colostrum.

Specific Comments:

  • Abstract:
    • Line 10, 59-60: Comments regarding pasteurized milk and colostrum occur several times, but they beg the question why pasteurized colostrum was not also evaluated (some bacterial species do survive the process). As a dairy scientist in the US, I do not know any farms that feed pasteurized colostrum. Collectively, these comments should be restricted to the discussion.
    • Line 23, 83, 285: I do not see the relevance of bacterial correlations to colostrum SCCs. SCCs are rarely taken and not considered from colostrum samples because of the cell/antibody/protein rich nature which is short lived and not indicative of mastitis or quarter health. Rather refractive index or antibody measurements are much more common. If aliquots of colostrum still exist these analyses should be added.

  • Summary?
    • Line 37: Is ‘implication’ the correct heading here?

  • Introduction:
    • Line 41: Cite previous study
    • Line 43-45: Overstepping language and should be revised. There is no evidence provided for calf supplementation, and no milk data shown.
    • Line 72: Should this be mastitis not mammitis?
  • Materials and Methods
    • Line 99-101: I’m not sure the information about cow care post calving is relevant but could be left in.
    • Line 103: This is very important. Were calves allowed to suckle before colostrum samples were collected?
    • Line 118, 123: -80 will be a freezer not a refrigerator.
    • Line 119: What preservative? Did these samples have to be diluted? Colostrum samples are often think/clumps/problematic for automated machine reading

  • Results:
    • Figure 1: Fix Y axis title on Shannon curve. Add (A) and (B) labels to the two panels and reference accordingly. If significantly different add ‘*’ symbol to graph and figure legend.
    • Line 183-184, 273: Consider rewording from “grouped left/right” to language like ‘clustered’
    • Figure 4: Image is of too poor resolution to evaluate.
    • Line 243: Data not showed= data not shown
    • Line 249: showed=shown
    • Line 249: Consider rewording to “abundance of certain bacteria correlated amongst genera present including...”
    • Line 282: Remove this sentence.
  • Discussion:
    • Line 305-307: Pasteurization comment is appropriate and well placed here.
    • Line 325-328: If this was a central objective this should be better articulated in the introduction. I think the results presented show reasonable stability and conservation although more farms across geological regions should be addressed.
    • Line 338-340: Cite previous studies
    • Line 360-362, 366: Check language and tense on this sentence. Also, use ‘mammary gland’ instead of ‘breast’.
    • Line 379:Reword ‘natural immune vaccination’
    • Line 402: microbe = microbes
    • Line 424: See comments above about SCCs. Additionally it was already known these animals were mastitis free, so relevance of SCCs falls again.

Reviewer 3 Report

Dear authors,

Interesting paper, very few comments, actually at the References presentation only, see nr. 4 and 33.

The researchers compared the bacterial composition of colostrum, milk and rectal samples and right through that you have a discussion about yes or no probiotics. Above that the section "Results" contains some content that should be part of M&M and some interpretations are presented there as "only a few ...", that is interpretation of results and therefor part of the discussion.

Some other comments:

-line 68: as far as I know, in the Western countries, pasteurization of colostrum or milk is not applied

-line 81-83: I see your point, but such a statement that bacterial community in colostrum ...   by diet composition and management, needs a ref.

-line 109: recycled bedding material, what about the microbiome of that??

-line 201-202:  that should not be part of the results, better in M&M, or discussion

-line 230, 236 etc. remove only, that is interpretation and Results must be sec Results

-line 311-321: here you introduce the use of probiotics, but this study is to my opinion not suitable for that. Therefor you need a case control model

-line 335: I do not see that at the end of your introduction at the objectives

-line 380-384: this is much more complicated than that, please skip this part.

-line 411-413: that is not correct, you have maximal found a correlation between ...

Reviewer 4 Report

Dear authors,

I read yur article, and have the following comments for you to consider:

Title: "Detection of the core bacreia in colostrum by analyzing colostral bacterial structure of two dairy farms and its association with that in feces and milk composition"
- consider re-writing the title: "bacreia" is probably "bacteria"; I also suggest "Detection of the core bacteria in colostrum and their association with the fecal/rectal microbiota and with milk composition in two dairy cow farms"

Introduction
------------
L72: mammitis? I think you wanted to write mastitis?
L78: I suggest not to use the term "microflora" and to replace it with "microbiota"
L78: I think you should also include other studies (more recent) on the composition of the colostrum microbiota for this comparison and for the discussion of your results (I can suggest "A  Randomized Controlled Trial of Teat-Sealant and Antibiotic Dry-Cow Treatments for Mastitis Prevention Shows Similar Effect on the Healthy Milk Microbiome", Frontiers in Vet. Sci., 2020, which is in Holstein Friesians)

Material & Methods
------------------
L138: which version of Qiime was used?
L139: I believe that the correct citation for Qiime is Caporaso et al. 2010, Nature)
L144-145: it's UniFrac distances that can be weighted or not, not the principal component analysis: please rephrase

Results
-------
L167: did you filter the OTU table for minimum number of counts and/or samples?

Round 2

Reviewer 1 Report

I thank the authors for their revised version of the manuscript. I have no further comments. 

Author Response

Thank you very much for your careful ans useful comments!

Reviewer 4 Report

Dear authors,

I acknowledge that your paper has improved after revision. I now have only a few minor aspects for your consideration:

  • check English spelling and typos (for example, L67: compositionin, space missing, etc.)
  • in your response to my previous comments, you say that you filtered the OTU table for minimum number of counts and/or samples: please write this also in the paper, and specify the thresholds that you used (minimum n. of total counts, or samples with counts, to retain OTUs in the table)
  • Please replace the word flora with the word microbiota throughout the text of the paper

Author Response

Dear reviewer,

  Thank you very much for your careful comment. We revised the manuscript again according to your comment,  changed 'flora' to 'microbiota'. I  am sorry for misunderstanding OTU filter process. We asked the engineer about this and knew that  we didn't filter the OTU table.

  With best regards!

 XiuRong Xu